

# Constraining the Accuracy of Flux Estimates Using OTM 33A

Rachel Edie[1], Anna M. Robertson[1], Robert A. Field[1], Jeffrey Soltis[1], Dustin A. Snare[2],
Daniel Zimmerle[3], Clay S. Bell[3], Timothy L. Vaughn[3], and Shane M. Murphy[1]

[1]University of Wyoming 1000 E. University Ave. Laramie, WY 82070
[2]All4 Inc., Kimberton, Pennsylvania 19442, United States
[3]Colorado State University Energy Institute, 430 N College Ave. Fort Collins, CO 80524

**Correspondence:** Shane M. Murphy (shane.murphy@uwyo.edu)

**Abstract.**

Other Test Method 33A (OTM 33A) is a near-source flux measurement method developed by the Environmental Protection Agency (EPA) primarily used to locate and estimate emission fluxes of methane from oil and gas (O&G) production facilities without requiring site access. A recent nation-wide estimate of methane emissions from O&G production included a large

number of flux measurements of upstream O&G facilities made using OTM 33A and concluded the EPA National Emission Inventory underestimates this sector by a factor of ~2.1 (Alvarez et al., 2018). The study presented here investigates the accuracy of OTM 33A through a series of test releases performed at the Methane Emissions Technology Evaluation Center (METEC), a facility designed to allow quantified amounts of natural gas to be released from decommissioned O&G equipment to simulate emissions from real facilities (Fig. 1). This study includes test releases from single and multiple points, from

equipment locations at different heights, and spanned methane release rates ranging from 0.16 to 2.15 kg h[-1]. Approximately 95% of individual measurements (N=45) fell within ± 70% of the known release rate. A simple linear regression of OTM 33A versus known release rates at the METEC site gives an average slope of 0.96 with 95% CI (0.66,1.28), suggesting that an ensemble of OTM 33A measurements may have a small but statistically insignificant low bias.

## 1   Introduction

Methane is a potent greenhouse gas, and emissions from the oil and gas (O&G) sector are thought to account for roughly 30% of total methane emissions in the United States (U.S. EPA, 2019). "Upstream" O&G activities (extraction, production, etc.) are thought to contribute the bulk of emissions within the O&G sector (Alvarez et al., 2012; Zavala-Araiza et al., 2015; Alvarez et al., 2018). However, attempts to quantify O&G methane emissions are hindered by inaccurate emission inventories, a lack of measurements, and variability between basins (Allen, 2016; Schwietzke et al., 2017; Robertson et al., 2017; Alvarez et al.,

2018; Omara et al., 2018). For example, basin-wide aircraft measurements of methane emissions from different O&G basins find emissions are generally higher than official inventories published by the U.S. Environmental Protection Agency (EPA) (e.g., Karion et al., 2013; Pétron et al., 2014; Karion et al., 2015; Schwietzke et al., 2017; Peischl et al., 2018), but the scale of aircraft measurements give little insight into the exact source of emissions on the ground. Site-level measurements are therefore necessary for improving emission estimates of the O&G production sector. Existing near-source studies of O&G basins suggest



the majority of large, uncontrolled emissions are the result of faulty equipment that may not be noticed for some time (Zavala-Araiza et al., 2017; Omara et al., 2018), emphasizing the need for permanent or semi-permanent monitoring technologies instead of infrequent manual inspections (Coburn et al., 2018; van Kessel et al., 2018). However, more permanent approaches are still under development and must be approved as equivalent monitoring technologies before they can replace existing EPA-

approved Leak Detection and Repair (LDAR) methods like optical gas imaging (OGI). Additionally, annual or semi-annual LDAR programs already in place rarely quantify total emissions from a site, and the efficacy of these programs depends on many factors including employee experience, leak size, and meteorological variables like wind speed and temperature (Ravikumar et al., 2016, 2018). This makes LDAR programs an important tool for reducing emissions, but they often do not provide data of the actual leak rate from production sites and this limits usefulness for improving emission inventories. In the

absence of OGI-equivalent continuous monitoring approaches, both basin and site-level emission estimates have been gathered using a number of different techniques, all with strengths and weaknesses.

One approach is to measure emissions onsite of an O&G production facility. Onsite measurement teams typically detect emissions from malfunctioning components via OGI, which can be quantified using high-volume samplers. Drawbacks of onsite measurements include difficulty measuring volatile organic compound (VOC) rich emission sources, inability to reach

all emission sources (such as the tops of free-standing tanks), difficulty measuring intermittent sources, and the time required for each inspection (Brantley et al., 2014; Bell et al., 2017; Ravikumar et al., 2018). Site access requirements also introduce the possibility of changes in operation when measurement teams are onsite (Alvarez et al., 2018).

The tracer flux ratio (TFR) technique estimates methane emissions by multiplying the observed concentration ratio of methane to a tracer by the known emission rate of the tracer. TFR has been used in both ground-based (Roscioli et al., 2015;

Yacovitch et al., 2015) and airborne applications (Daube et al., 2018), though only ground-based approaches have been used for O&G facilities. TFR can quantify all emissions at an O&G facility, and can often differentiate between emissions from relatively close facilities without the need for site access, though access can improve flux estimates. TFR does not require an atmospheric transport model and is therefore insensitive to uncertainties in atmospheric stability and turbulence. Drawbacks of this technique include the reliance on downwind roadways of sufficient distance ($\sim$0.5 - 2 km), reliable wind direction, the

amount of time required to estimate emissions from one site (>1 hr), and the need to release tracer gases (typically acetylene or nitrous oxide) near O&G facilities.

As mentioned previously, airborne mass flux measurements have been used to estimate methane emissions from multiple O&G basins (eg., Karion et al., 2013, 2015; Peischl et al., 2015, 2016, 2018; Pétron et al., 2014; Schwietzke et al., 2017). Meteorological requirements (like a fully developed planetary boundary layer and consistent wind direction) make these mea-

surements difficult, especially for expansive O&G basins such as the Permian basin in Texas and New Mexico (Peischl et al., 2018). Emission estimates of individual production sites via aircraft measurements are also possible, but measurement sites typically need to have relatively large emissions and are limited by aircraft range, turning radius, and favorable meteorological conditions (Caulton et al., 2014; Lavoie et al., 2015, 2017; Conley et al., 2017). Additionally, airborne sampling must occur during the day to meet meteorological requirements, and diurnal variability of emissions associated with onsite maintenance





could impact aircraft-based emission estimates in some basins (Schwietzke et al., 2017; Vaughn et al., 2018; Zaimes et al., 2019).

A final type of measurement technique used to estimate emissions from O&G production facilities—and the focus of this study—are downwind measurements that estimate emissions by using methane mixing ratio and wind measurements to derive

the source flux. Downwind emission flux estimates are made using parameters measured in the field combined with additional parameters found with Gaussian or atmospheric dispersion models (Brantley et al., 2014; Rella et al., 2015; Caulton et al., 2018; Robertson et al., 2017; Lan et al., 2015; Foster-Wittig et al., 2015). Downwind measurements do not require site access, but may not be able to identify or capture all sources onsite, especially buoyant ones. Similar to TFR, these techniques require downwind roadways (50 - 200 m away) and consistent wind direction. Operator-approved site access can improve OTM 33A

measurement success in regions with limited downwind roadway infrastructure or complex topography. Though sampling time can be considerably faster than TFR or onsite techniques, it is hard to measure enough sites to get a representative sample (and therefore a flux) of an entire O&G basin (Harriss et al., 2015). As a whole, all of the emission measurement techniques mentioned here are only representative of a timescale between seconds and hours, and therefore have difficulty capturing emissions sources with large temporal variability (U.S. EPA, 2014; Brantley et al., 2014; Robertson et al., 2017; Bell et al.,

2017; Caulton et al., 2018; Vaughn et al., 2018).

This study focuses on a ground-based mobile emissions measurement approach, Other Test Method 33A (OTM 33A). OTM 33A is among the most common downwind methods, along with TFR, used to measure methane and VOC fluxes from O&G sources (Brantley et al., 2014, 2015; Robertson et al., 2017). A recent study by Bell et al. (2017) compared onsite, OTM 33A, and TFR measurement techniques in the Fayetteville Shale. The results of the Bell et al. study suggest OTM 33A only captured

~40-60% of emissions measured or estimated by onsite teams in the Fayetteville when the dominant emission source was an onsite direct measurement rather than a simulated emission source. OTM 33A had a larger low bias when manual or automated unloadings were measured. Manual or automated unloadings occur when the well pressure is not great enough to move liquids from the geologic formation, preventing gas flow to the pressurized sales line. To maximize the pressure differential, the well is vented directly to the atmosphere in order to remove accumulated liquids. This process can be performed manually

or automatically, and may use a plunger to assist with liquid removal. This creates an emissions plume with high vertical velocity. It is likely the majority of this plume would pass over the mobile laboratory unless perfect conditions and road access generate a downwind measurement site 200 m or less from the source. The results of the Bell et al. study add uncertainty to recent national methane emission estimates, which relied heavily on OTM 33A measurements in five O&G basins (Alvarez et al., 2018). However, the Alvarez et al. study also found that basin-wide emission estimates based on OTM 33A facility

measurements agreed with airborne basin-wide flux estimates to within measurement uncertainty. Additionally, no significant low-bias (> 10%) was detected in numerous (>100) OTM 33A test releases, conducted by multiple groups (Brantley et al., 2014; Robertson et al., 2017). These test releases were all single point-source releases conducted in open terrain without obstacles, which may not be a reliable comparison to the types of emission sources experienced in O&G fields. The discrepancy between results of Bell et al. study and previous test releases, along with the potential significant impact on national emission estimates,

motivated the suite of more realistic test releases described here.





## 2 Materials and Methods

### 2.1 Mobile Laboratory

The University of Wyoming mobile laboratory is a customized Freightliner Sprinter van. The front of the van is equipped with a mast that projects instrumentation and the inlet 4 meters above the ground slightly beyond the vehicle's front bumper.

Meteorological instruments on the mast include a 3-D sonic anemometer and an all-in-one compact weather station. The mast also includes a camera, an AirMar differential GPS, and a Teflon inlet (1/4" OD) for gas-phase species. Ambient air is pulled through the Teflon inlet at a rate of 6.5 L min$^{-1}$. For the test releases described here, the laboratory was instrumented with a G2204 Picarro Cavity Ringdown Spectrometer (CRDS) which has been modified to measure water vapor and dry methane concentrations at a frequency of 2 Hz. The Picarro has an additional meter of 1/8" OD Teflon tubing that branches from the

main inlet line, resulting in a total sample transit time through the inlet to the instrument of one second. This lag is accounted for during data processing. Additionally, the van contains a battery bank which allows the instrumentation and data acquisitions system to be used while the vehicle engine is turned off.

### 2.2 Instrument Calibration

The Picarro response was tested using two NIST certified methane-zero air mixtures (2.538 $\pm$ 0.05 ppm, 101 $\pm$ 5 ppm), and

ultra-high-purity zero air (UHPA) at intervals throughout the campaign to confirm stability and accuracy. The instrument was always within $\pm$ 0.01 ppm of the lower NIST standard, $\pm$ 1 ppm of the higher standard, and $\pm$ 0.003 ppm of zero when tested with UHPA. The 5-second instrument precision is $\pm$ 0.002 ppm. Due to the observed instrument stability and accuracy, no calibration adjustments were made to methane concentrations during data processing.

### 2.3 OTM 33A Measurement Method

OTM 33A is one of the EPA Geospatial Measurement of Air Pollution Remote Emission Quantification (GMAP-REQ) techniques (U.S. EPA, 2014; Thoma, 2012; Brantley et al., 2014, 2015; Robertson et al., 2017). OTM 33A has two operational parts; first to detect and second to quantify emissions. Detection of emissions occurs by driving downwind of possible emission sources in an attempt to transect an emissions plume, measure the ambient background trace gas mixing ratio, and, if possible, to rule out any emissions from upwind sources. If enhancements of methane or other trace gases are detected during

downwind transects of a possible source, the laboratory is parked 20–200 m directly downwind within the emission plume to quantify emissions. Care is taken to orientate the mast directly into the dominant wind direction to minimize impact from turbulent eddies around the vehicle. Once the laboratory is safely positioned, the vehicle is turned off and an OTM 33A flux measurement begins. During the ~20 minute measurement, 2 Hz measurements of wind direction (in x, y, and z), wind speed, temperature, and the methane mixing ratio are collected and time-stamped with a universal data system time. Meanwhile, dis-

tance to the possible emission sources relative to the mast of the laboratory are measured using a TruePulse laser range finder



(Model 200). If possible, the most likely emission source is identified using an infrared camera (FLIR GF300). Site photos and observations are also collected.

The OTM 33A analysis program, written in MATLAB (2015), estimates an emission mass flux, Q [g s⁻¹], by using the Gaussian dispersion equation (Eq. 1). The terms of this equation are found as follows. First, the lowest 5% of measured mixing

ratios during the ∼20 minute measurement are averaged and considered ambient background, which was around 1.9 ppm (±0.15 ppm) of methane for this study. The background value is subtracted from the data to yield methane enhancement. The analysis program bins observed methane enhancements by wind direction into 10° bins (Fig. 2(a)), and then calculates the average methane enhancement observed in that wind bin. A plot of methane enhancement vs. wind direction is then generated and fit to a Gaussian distribution (Fig. 2(b)). The Gaussian fit's apex is $C_{peak}$ [g m⁻³]. To determine the expected spreading of

the emission plume, the program calculates atmospheric stability indicator values (ASI). The ASI are based on the standard deviation of the 2-dimensional wind direction (horizontal spreading), and the standard deviation in vertical wind speed (vertical spreading), also known as the turbulent intensity. The horizontal and vertical ASI are averaged together into a Point Gaussian Indicator (PGI) value, which parameterizes the vertical and horizontal plume spread experienced during the OTM measurement. There are seven PGI values which correspond to Pasquill stability classes A-D (Brantley et al., 2014; U.S. EPA, 2019). The

PGI and measured source distance are used as inputs to a lookup table that gives the plume dispersion in two dimensions, $\sigma_y$ [m] and $\sigma_z$ [m]. The average wind speed $\bar{U}$ [m s⁻¹] is also calculated for the same time periods methane enhancements are observed.

$$Q = 2 \times \pi \times \sigma_y \times \sigma_z \times \bar{U} \times C_{peak} \tag{1}$$

Equation 1 does not include any terms for ground reflection of the plume, plume buoyancy/velocity, or differences in height

of the emission source and measurement inlet. OTM 33A assumes a single emission point. For this reason, OTM 33A is best suited for measuring O&G facilities with equipment concentrated in one area that have downwind roadways. OTM 33A struggles to quantify plumes with a particularly high vertical velocity or buoyancy (like manual unloadings or very hot emissions). In this scenario, the calculated $C_{peak}$ will not represent the center of the emission plume, leading to underestimations of these sources (Bell et al., 2017).

A series of built-in data quality indicators (DQI) will flag an OTM 33A flux estimate for a variety of reasons, including poor Gaussian fit, inadequate sampling time within the emission plume, too variable wind speed or direction, or a maximum methane enhancement that is too small. Flags are then added up, and measurements are broken into categories that represent the probability an OTM measurement is a good flux estimate. For the current study, the same approach as Robertson et al. (2017) and Bell et al. (2017) was used where most of the Category 1 and a few Category 2 measurements that were only

flagged for low methane concentrations were considered. Occasionally, measurements with very few DQI flags (Category 1 measurements) will be thrown out after review of the Gaussian fit or if IR camera images suggest we are missing most of the emission plume. Full descriptions of the DQI can be found in SI Sect. 1.2, Robertson et al. (2017), Brantley et al. (2014), and in the EPA's documentation (U.S. EPA, 2014).



## 3 Results

### 3.1 Test Releases

The University of Wyoming performed two sets of test releases to assess the ability of OTM 33A to quantify methane emissions. The first set of tests, the Christman Field Test Releases (CF-TR) were conducted in conjunction with Colorado State

University in July and August of 2014 at the abandoned Christman Airfield in Fort Collins, CO. These releases consisted of two configurations, a simple point source (an opened gas cylinder) and manifold (an elevated ∼6-foot length of PVC pipe with many perforations). Neither source of methane gas was obstructed, and they were, in essence, single point sources, one slightly broader than the other. Release rates were set using calibrated mass flow controllers and are correct to within 5%. These tests spanned a variety of release rates (0.2 to 2 kg hr$^{-1}$) and were staged in an open field with no obstructions (clear line of site)

between the single methane source and mobile lab. Winds ranged from 2–8 m s$^{-1}$ from the S/SE. The calculated PGI ranged from 2–6, which roughly correspond to Pasquill-Gifford stability classes A–D. Details of these results are reported in Snare (2015) and Robertson et al. (2017).

The more-recent set of tests were performed at the Methane Emissions Technology Evaluation Center (METEC) in Fort Collins, CO in June of 2017. METEC contains multiple faux O&G facilities ranging in size and complexity with decommis-

sioned O&G equipment that has been plumbed to release a known amount of natural gas (>94% methane) from a multitude of points. For this study, we used one METEC site representative of a small O&G facility that included a condensate storage tank, separator, and well head, all of which were plumbed to be possible emission sources (Fig. 1). This resulted in release configurations that had multiple release points at different heights (0–3 meters), and up to 6 meters apart from one another. The relative complexity of the site also introduced obstructions (the methane release would have to flow around a large tank

or other piece of equipment to reach the mobile lab) which could potentially impact release quantification. Releases spanned 0.17 to 2.15 kg hr$^{-1}$ and were controlled by combining flows from a number of critical orifices, resulting in a four $\sigma$ release error less than 5%. Meteorological conditions ranged from sunny to partly cloudy, with average winds from 2–9 m s$^{-1}$ from the E/SE. The calculated PGI ranged from 3–6, which roughly correspond to Pasquill-Gifford stability class B–D.

23 OTM 33A test releases were measured during the CF-TR; 21 passed the data quality indicators (DQI) (U.S. EPA, 2014)

25 and were included in this analysis. 34 test releases were measured during the METEC-TR, of which 24 passed the DQI and were included in this study. A similar success rate, ∼70%, has been observed in the majority of the basins measured by the University of Wyoming (Robertson et al., 2017). The following analysis explores different statistical approaches to constrain the error associated with individual OTM 33A measurements and to assess the accuracy and precision of an ensemble of OTM 33A measurements. The latter analysis is especially important given this is how OTM 33A measurements are often scaled up

30 to estimate basin-wide emissions from O&G.



### 3.2 Evaluating the Accuracy of OTM 33A

#### 3.2.1 Percent Error Analysis

$$\% \, Error = \frac{OTM \, 33A \, flux - known \, release}{known \, release} \times 100 \tag{2}$$

Percent error, Equation 2, was calculated for each measurement made during the test releases. A histogram of percent error

for both the CF-TR and METEC-TR indicate a large range in over- and under-estimations are possible using OTM 33A (Fig. 3). Percent error ranges from -75% to 50% and -60% to 170% for CF-TR and METEC-TR respectively. Figures 4 and 5 show that the larger % errors correlate with smaller release rates, with OTM 33A generally overestimating smaller releases. 68% of the CF-TR data fall within $\pm 28\%$ of the known release, which is the $1\sigma$ error used by Robertson et al. and similar to the error reported by the EPA of 72% of measurements within $\pm 30\%$ of the known release (Brantley et al., 2014). 68%

of the METEC-TR are within $\pm 38\%$ of the known release, perhaps suggesting that a slightly higher $1\sigma$ error is appropriate, especially if measuring emissions fluxes less than 0.5 kg hr$^{-1}$. For the combined set of test releases, greater than 85% of the data are within $\pm 50\%$ of the known value, and 95% of the data are within $\pm 73\%$. If a Gaussian curve is fit to all of the test release data (N=45), the 95% confidence interval is found to be $+54\%$ to $-84\%$, suggesting a low bias of $-15\%$ and a $2\sigma$ error of $\pm 69\%$ (Fig. S5). The rounded $2\sigma$ confidence interval for test releases of $\pm 70\%$ would become $0.58q$ and $3.33q$ when $q$ is an

OTM 33A estimate made of an unknown emission source in an O&G basin.

#### 3.2.2 Ordinary Least Squares Regression

Another approach to assess the performance of OTM 33A is using an ordinary least squares (OLS) regression applied to a correlation plot of the OTM 33A flux estimate versus the known release rate. Assuming the OTM-measured flux and known release rate converge at (0,0) yields OLS slopes of 0.91 for CF-TR and 0.92 for METEC-TR (Fig. 6). This suggests OTM 33A

may have a $\sim -10\%$ negative bias when an ensemble of measurements are considered. Notably, the increased complexity of the METEC-TR did not yield a more significant bias like that reported by Bell et al. 2017. Statistical analysis of the residuals to assess point leverage and possible outliers supports the validity of an OLS approach for the test release data. Removing the largest outlier found with the Cook's test improves both OLS fits slightly to 0.97 for METEC-TR and 1.1 for CF-TR. Residual plots are included in SI Sect. 2.

95% confidence intervals (CI) for the OLS fit were calculated through bootstrapping following the method detailed in Robertson et al. 2017, whereby the linear regressions of bootstrapped data sets are calculated to assess the range of possible regressions. Bootstrapping was used because it does not require an assumption of normally distributed data (unlike the Gaussian fit approach used in Sect. 3.2.1). Using this method, the OLS correlation slopes have a mean (and 95% CI) for the CF-TR of 0.96 (0.56,1.47) and 0.96 (0.66,1.28) for the METEC-TR.



### 3.2.3 Bland-Altman Analysis

Because the rate of methane releases for both the METEC and Christman tests are known to within a small margin of error (<5%), OLS regression, which assumes no error in the independent variable, is a reasonable approach. However, OLS analysis is weighted by larger release rates and may not give an accurate representation of OTM 33A performance at all methane

emission rates. Bland-Altman (BA) analysis removes this bias by considering the difference between the test release and OTM measurements (known release - OTM flux) as a function of a known release rate (Fig. 7) (Giavarina, 2015). Bland-Altman analysis also assumes that the method difference (y-axis) comes from a normal distribution. Kolmogorov-Smirnov statistical tests supporting the normality of the method difference can be found in SI Sect. 3. For BA analysis, if the $2\sigma$ range of method difference includes zero, the methods are considered to be statistically equivalent; i.e. no bias (Giavarina, 2015). The BA

plot also illustrates the amount by which OTM can over- (negative numbers) or under-estimate (positive numbers) the known release. On average, the CF-TR and METEC-TR both underestimate the known releases, with mean differences of 0.028 kg h⁻¹ and 0.025 kg h⁻¹ respectively. However, since the $2\sigma$ interval includes zero, the BA analysis identifies no statistical difference between the OTM 33A flux estimate and the known release rate.

### 3.2.4 Orthogonal Distance and Variance Weighted Least-Squares Regression

Other approaches for minimizing the influence of larger release rates on the OLS fit include orthogonal distance regression (ODR) and variance weighted least squares regression (VWLS). These methods take into account error in both the x and y variables, and require that each measurement has an independent uncertainty estimate on both axes. Since uncertainty of OTM 33A flux estimates is taken as a fixed percentage of the estimated value, these methods tend to perceive a higher confidence (smaller absolute uncertainty) in smaller estimates, and a lower confidence (higher absolute uncertainty) in larger estimates.

This results in a fit with a low bias since the estimates with smaller absolute uncertainty are strongly weighted, and less weight is given to estimates with larger uncertainties. To examine these approaches, the METEC-TR are used as an example below.

Applying a measurement uncertainty of ±50% (representing the % error that roughly 85% of the data points are within) for each OTM measurement and the metered uncertainty for each METEC-TR in kg h⁻¹ yields an ODR slope of 0.79 ±0.09 when the intercept is set to (0,0) (Fig. 8). A lower slope of 0.67 ± 0.1 is found using the VWLS method. In this case the ODR

and VWLS regressions suggest the OTM flux estimates are 20–33% lower than the known releases, where an OLS regression indicates the method is only 8% low. Total emissions estimated by OTM 33A (23.074 kg hr⁻¹) are 2.5% lower than the total known emission rates (23.67 kg hr⁻¹), suggesting OLS regression is a better fit for this data set.

VWLS and ODR should be used with caution where the measurement uncertainty is not independent of the measurement (i.e. a constant fractional error), because this may discriminate against data points of a larger magnitude depending on uncertainty

in the other (x or y) variable. Bell et al. (2017) used a VWLS regression to compare OTM 33A measurements to onsite measurements of O&G production facilities in the Fayetteville, which yields a correlation of 0.41 (+0.51, −0.17). The 95% CI of the VWLS regression are calculated through bootstrapping the regression and considering the uncertainties in both the study onsite estimate and the OTM 33A estimate for each data point. Repeating this analysis using an OLS regression without an





intercept results in a correlation of 0.39 (+0.39, -0.15) (Fig. S6). If an OLS regression were chosen instead of the VWLS, the conclusion by Bell et al. 2017 that OTM 33A produced lower emissions estimates than the onsite measurement results would not have been effected. The total mass flux measured by OTM 33A (13 (+5.3, −2.1) kg h$^{-1}$) and onsite teams (19 (+7.7, −3) kg h$^{-1}$) also supports the conclusion that the OTM 33A flux estimate was biased low relative to the onsite measurements at

these paired facilities.

### 3.3   Sensitivity Analysis

#### 3.3.1   OTM 33A Sensitivity to Source Distance

Because OTM 33A assumes a point source, the distance to the release point has a large influence, as this impacts the modeled plume spread and therefore the final calculated flux. The importance of an accurate source distance in Gaussian plume modeling

has been noted in previous studies (Lan et al., 2015; Caulton et al., 2018). During the METEC-TR, the University of Wyoming measurement team had site access and we were able to determine the exact emission point(s) using an IR camera. With this knowledge, we were able to calculate the exact source distance, or the average distance in the case of multiple emission sources. In the field, site access is often not available and it's often not possible to detect the most likely emission point(s). For this reason, the average distance of possible emission points is used when calculating source distance.

For a 5% change in source distance, the OTM 33A flux estimate increased by almost 10% (Fig. 9(a)). Although the well pad measured at the METEC facility was quite small, the average source distance was larger than the specific source distance ∼60% of the time. Source distance related error is small in the context of the ±70% measurement error, but underscores how determination of the exact emission point can further reduce errors in the field. Looking at this change in terms of a kg hr$^{-1}$ flux, Fig. 9(b) shows that the OTM flux estimated by the average or specific source distance has very little impact in the over- or

under-estimation of the METEC known release. Allowing this fit to have an intercept changes the linear fit to $y = 0.978x - 0.03$, a negligible difference.

#### 3.3.2   OTM 33A Error - Sensitivity to Wind Speed

One hypothesized reason for the underestimation of OTM 33A compared to onsite methods reported in the Bell et al. study is the lower wind speeds (<2 m s$^{-1}$) experienced in that study. The CF-TR and METEC-TR both had wind speeds higher than

2 m s$^{-1}$, making an absolute conclusion impossible, but for the wind speeds measured there is no obvious trend between the mean measurement wind speed and OTM 33A error (Fig. 10).

#### 3.3.3   OTM 33A Error - Sensitivity to Number of Sources and Source Height

The METEC-TR included multiple emission points, both slightly above and below the sampling inlet height. There is no obvious trend between the number of release points and % error, though the sample size for two or more sources is relatively

small. The height of the source also shows no obvious influence on OTM 33A accuracy.





### 3.4 Ensemble Mass Flux

OTM 33A measurements are often used to find a average emission rate per well or per facility in an O&G basin (Robertson et al., 2017; Alvarez et al., 2018). To assess the accuracy of the mean of a number of OTM 33A measurements, the mean mass flux measured by OTM 33A is compared to the mean mass flux of the known release through bootstrapping. The bootstrapping

approach is used to generate more statistically robust results without the need for assuming Gaussian distributions. The OTM 33A flux estimates and known releases (including their respective measurement uncertainties) are sampled with replacement, summed, and compared following Robertson et al. (2017). This approach suggests that the addition of complexity in the METEC-TR did not significantly impact the accuracy of OTM 33A (Fig. 12), and for both sets of test releases there is a large amount of overlap between the OTM 33A and known release distributions (Fig. S6). These results also indicate OTM 33A

does not drastically underestimate the total emissions for an ensemble or group of measurements, and that scaling-up mean emissions measured with OTM 33A to an entire basin is a valid approach.

Figure 13 summarizes measurement conditions experienced during four University of Wyoming field campaigns (Robertson et al., 2017) and the test releases. Data from the four O&G basins were used by Alvarez et al. (2018) to generate an estimate of national methane emissions. Measurement conditions in Fayetteville, Arkansas (AR) were notable for closer source distance,

lower wind speeds, and generally more unstable atmospheric conditions. All of these variables could have influenced the low bias reported by Bell et al. and are conditions that should be replicated (if possible) in future test releases.

### 4   Conclusions

The more realistic test releases described in this study build on preexisting test releases and suggest a single OTM 33A measurement can have a $2\sigma$ error of $\pm70\%$. Analysis of both the simple CF-TR and more complex METEC-TR indicate that under

these measurement conditions and release rates, an ensemble of OTM 33A may have a slight negative bias ($\sim5\%$) when compared to a known release rate through an OLS model. The mean and 95% CI found through bootstrapping are 0.96 (0.56,1.47) and 0.96 (0.66, 1.28) for the CF-TR and METEC-TR respectively. The 40-60% underestimation reported in the Bell et al. study was not replicated during either test release experiment.

OTM 33A flux estimates are sensitive to the assumed source distance, with a +5% change in source distance corresponding

to a $\sim+10\%$ change in the OTM flux. However, the error caused by uncertainty in source distance is small compared to the measurement method error determined through these test releases. During field measurements, uncertainty in source distance can be mitigated by having site access and an IR camera to detect the emission source(s). Uncertainty did not correspond to wind speeds observed during the test releases, but was relatively higher for smaller release rates. Sensitivity of OTM 33A to the number or height of emission sources was inconclusive.

OTM 33A has been used to estimate mean facility emissions and basin-wide facility emissions in a number of O&G basins. The mean mass fluxes and 95% CI for each test release experiment are not statistically different. This analysis lends confidence to nation wide emission estimates from the O&G production sector using OTM 33A measurements. The analyses reported here and the study by Bell et al. suggest that OTM 33A does not overestimate an ensemble of flux estimates.





*Code and data availability.* Available on request.

*Author contributions.* Authors RE, ARM, DAS, JS, SMM, and RAF collected data and helped design the study. Authors DZ, CSB, TLV helped with study design, statistical methods, and study comparison. Authors RE, AMR, SMM, RAF, and CSB wrote and edited the manuscript. Authors DAS, JS, and DZ provided feedback on the manuscript.

5   *Competing interests.* The authors have no competing interests to report.

*Acknowledgements.* The authors acknowledge support from the Wyoming School of Energy Resources Center of Excellence in Air Quality, the Clean Air Task Force, and the Methane Emissions Technology Evaluation Center.





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




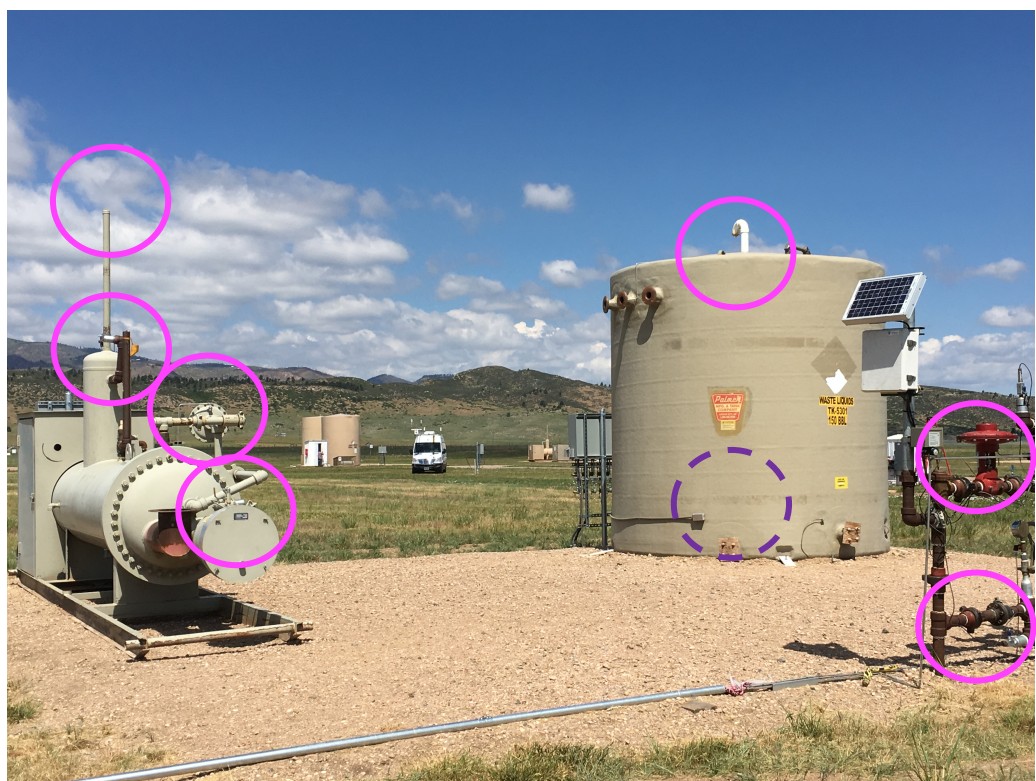

**Figure 1.** METEC facility with possible release points circled. UW mobile laboratory in background.





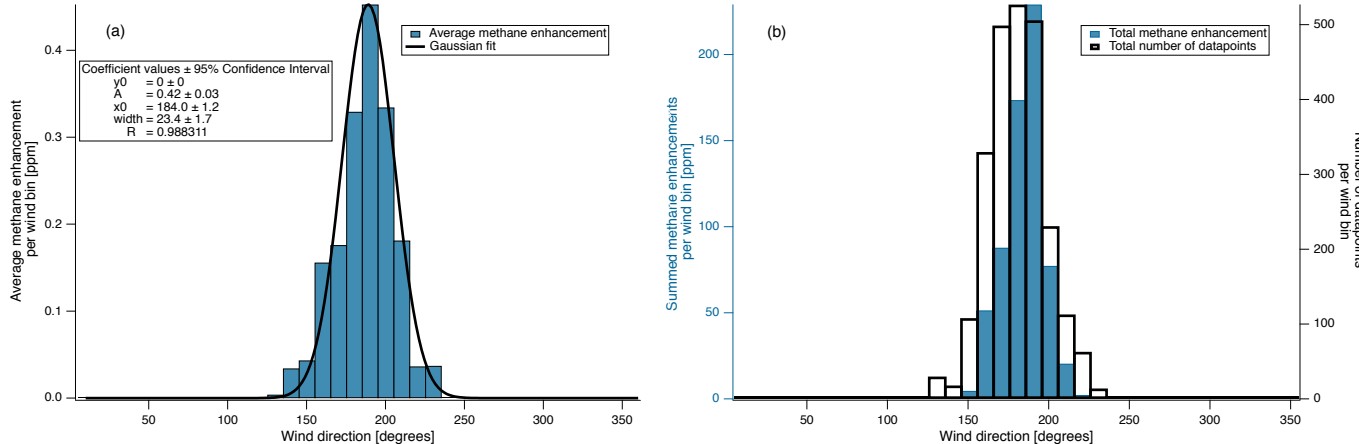

**Figure 2.** Summed methane enhancement and total number of datapoints in each 10°wind bin (a). Average methane enhancement per 10° wind bin and Gaussian fit (b). Goodness of fit parameter R is calculated following Eq. (S1).

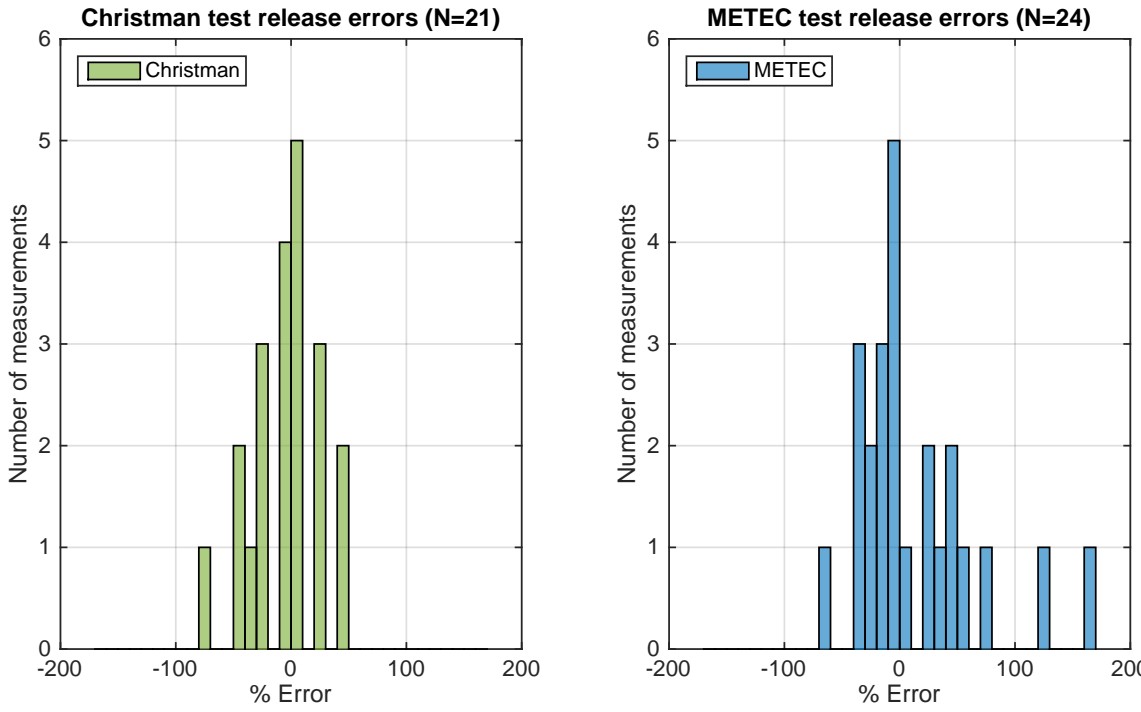

**Figure 3.** Histogram of percent error of the OTM 33A flux estimate for both Christman and METEC test releases. Data are binned in 10% error bins. Positive % error corresponds to OTM 33A over-estimating the known release rate.



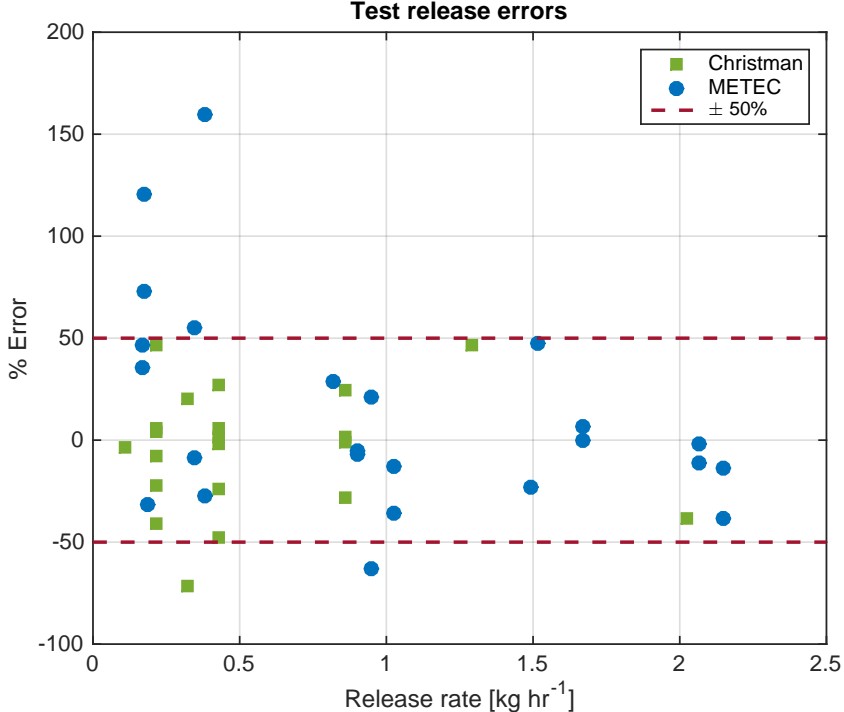

**Figure 4.** Scatter plot of test release error and release rate. Positive % error corresponds to OTM 33A over-estimating the known release rate.



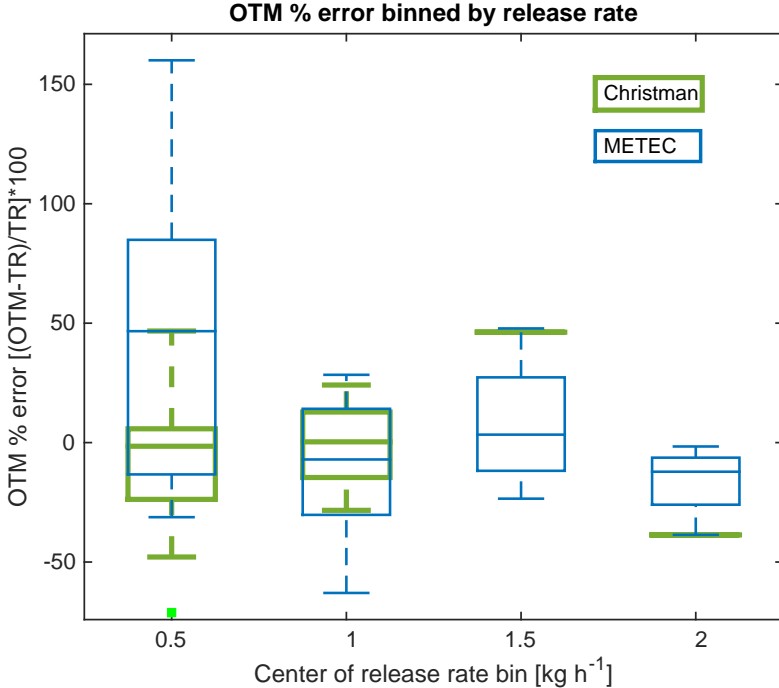

**Figure 5.** Box plot of METEC and Christman OTM 33A release errors binned by release rate. The rectangle contains the median value, while the edges represent the 25[th] and 75[th] percentiles. Box whiskers include the rest of the data (100% coverage). Positive % error corresponds to OTM 33A over-estimating the known release rate.



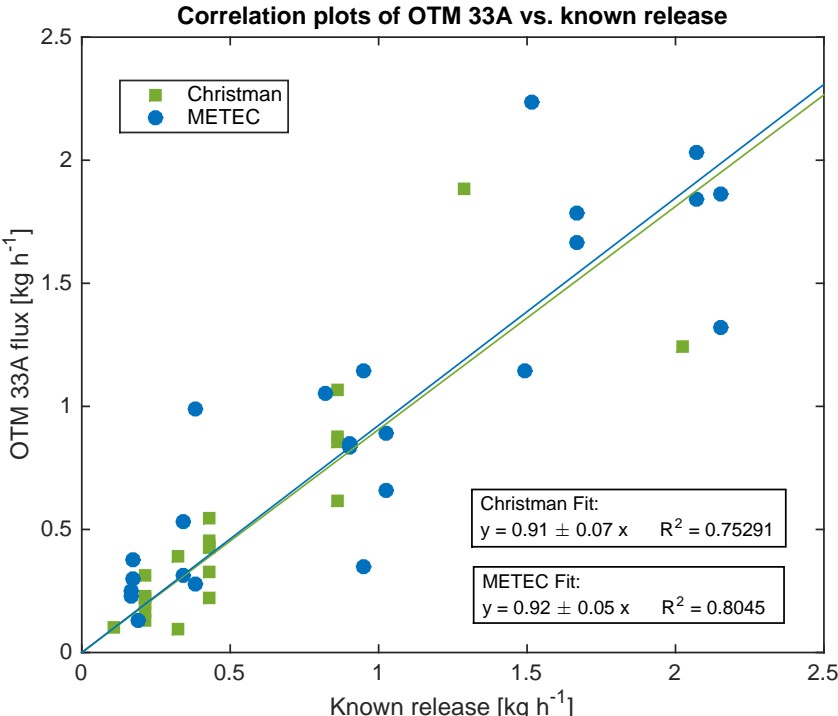

**Figure 6.** Correlation plot of OTM 33A-measured flux versus known release rates. Intercept is set to (0,0).





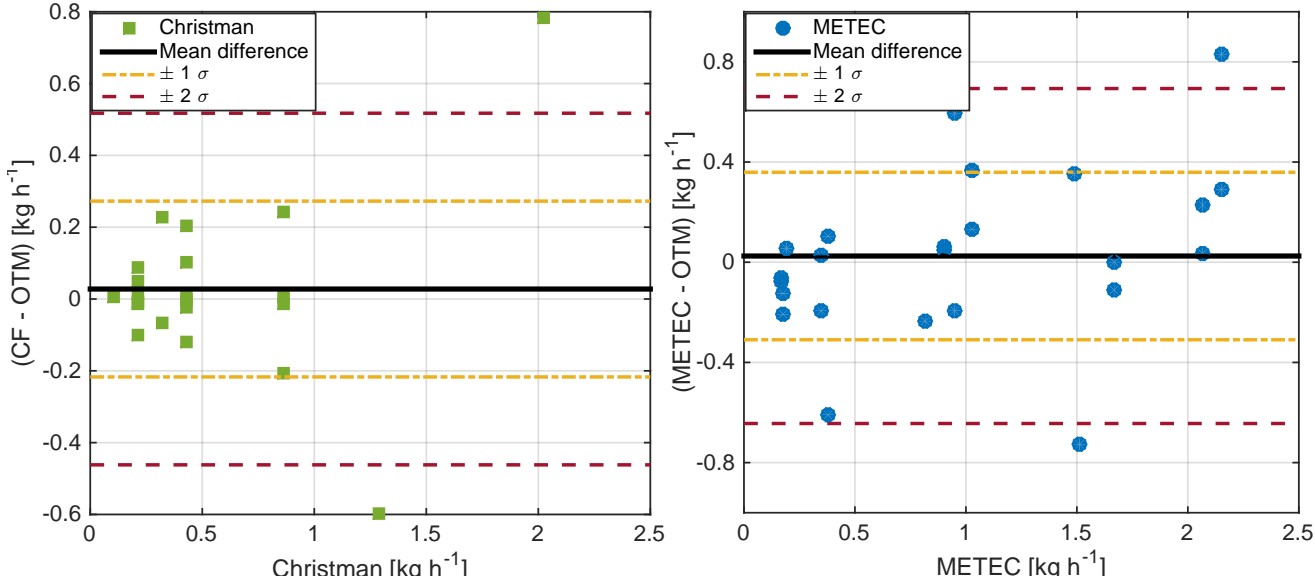

**Figure 7.** Bland-Altman analysis of Christman and METEC test releases. Bold black lines represent the mean difference between the test release and the OTM measurement, while the dashed yellow and red lines indicate one and two standard deviations of the method difference.



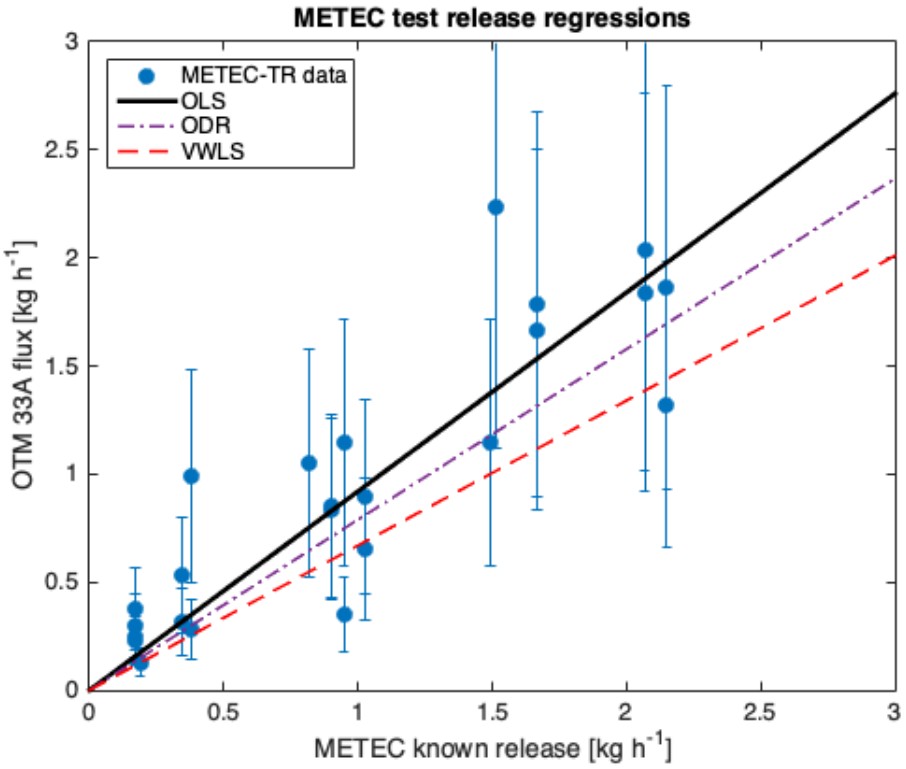

**Figure 8.** Correlation plot of known release and OTM 33A flux estimate for the METEC test releases. Plot includes OLS (slope = 0.92), ODR (slope = 0.79), and VWLS (slope = 0.67) regressions.



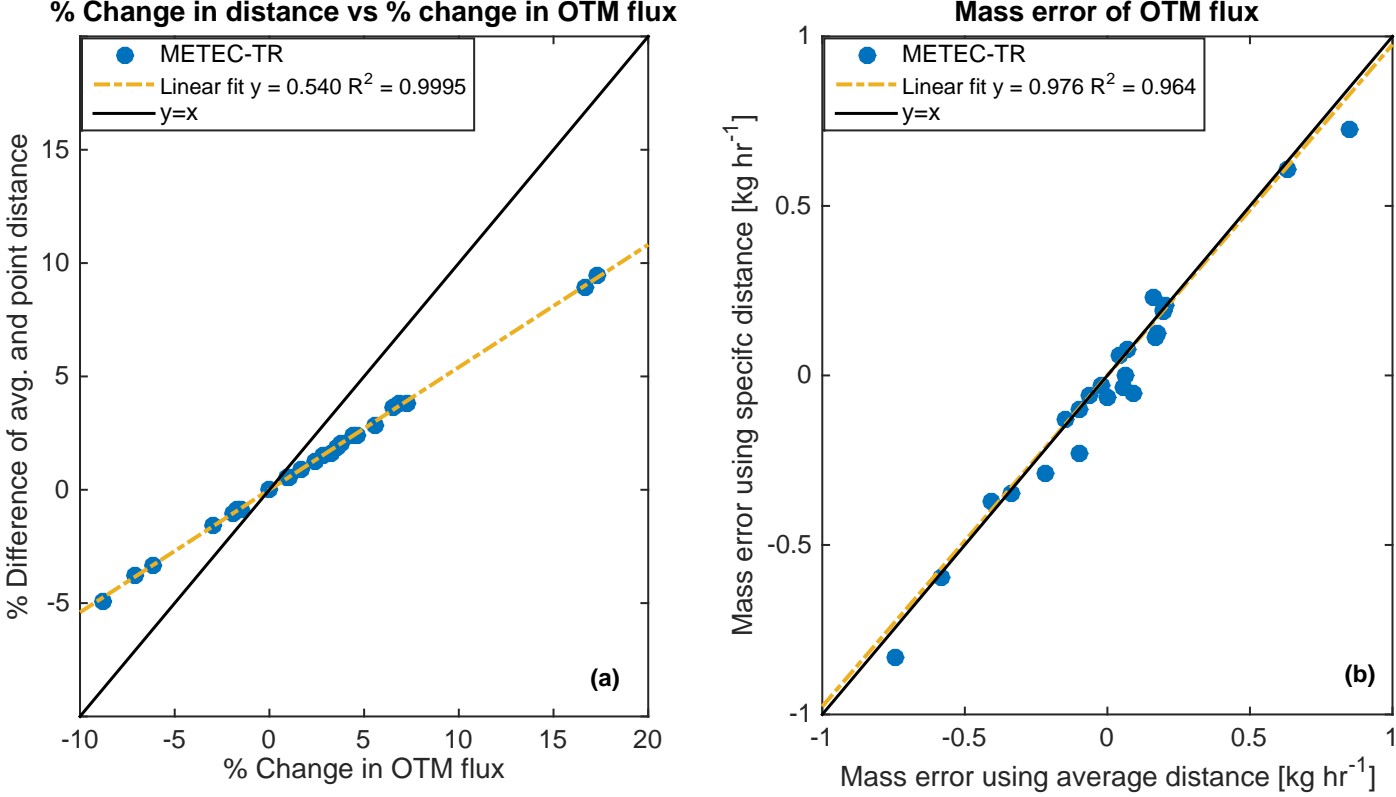

**Figure 9.** (a) Correlation between percent change in distance ((Avg. distance - point)/point * 100), and resulting percent change in OTM flux ((OTM avg - OTM)/OTM * 100). (b) OTM 33A flux mass error compared to the METEC known release when using average versus known source distance.



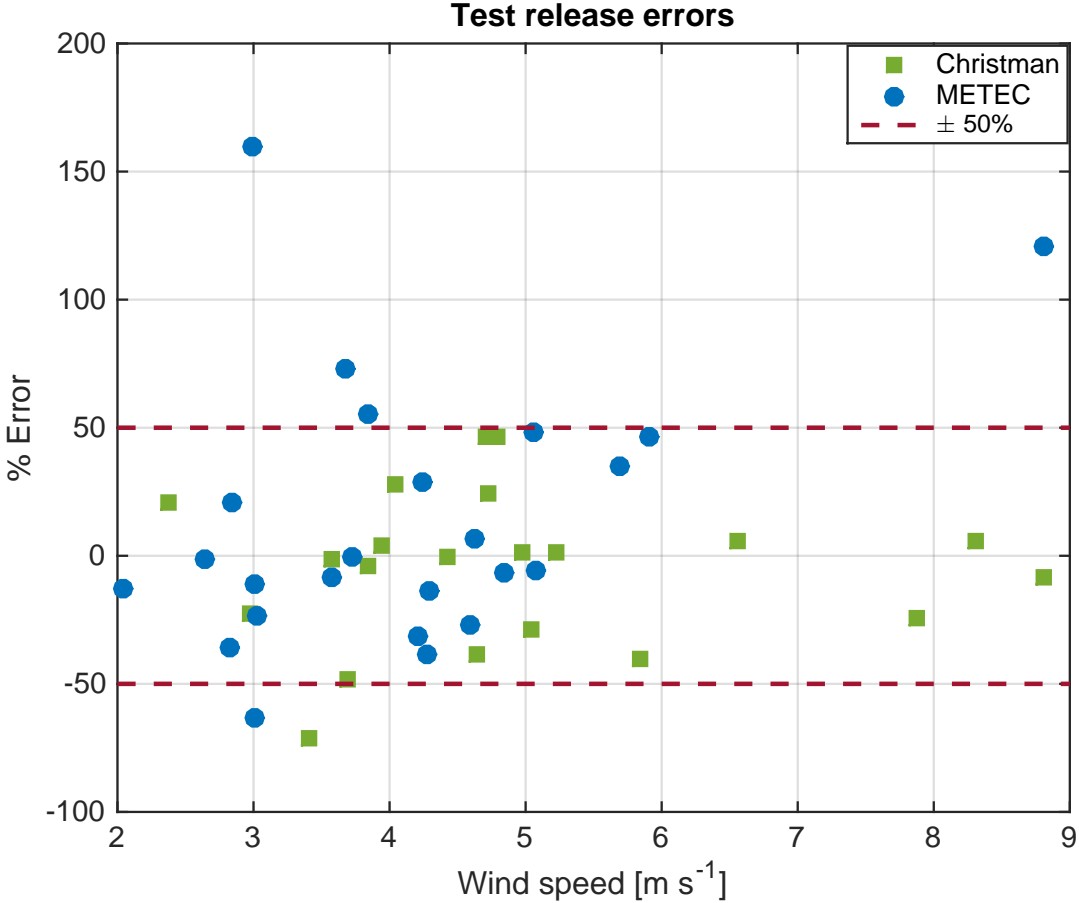

**Figure 10.** OTM 33A percent error compared to the mean measurement wind speed. Positive % error is OTM overestimating the known release.

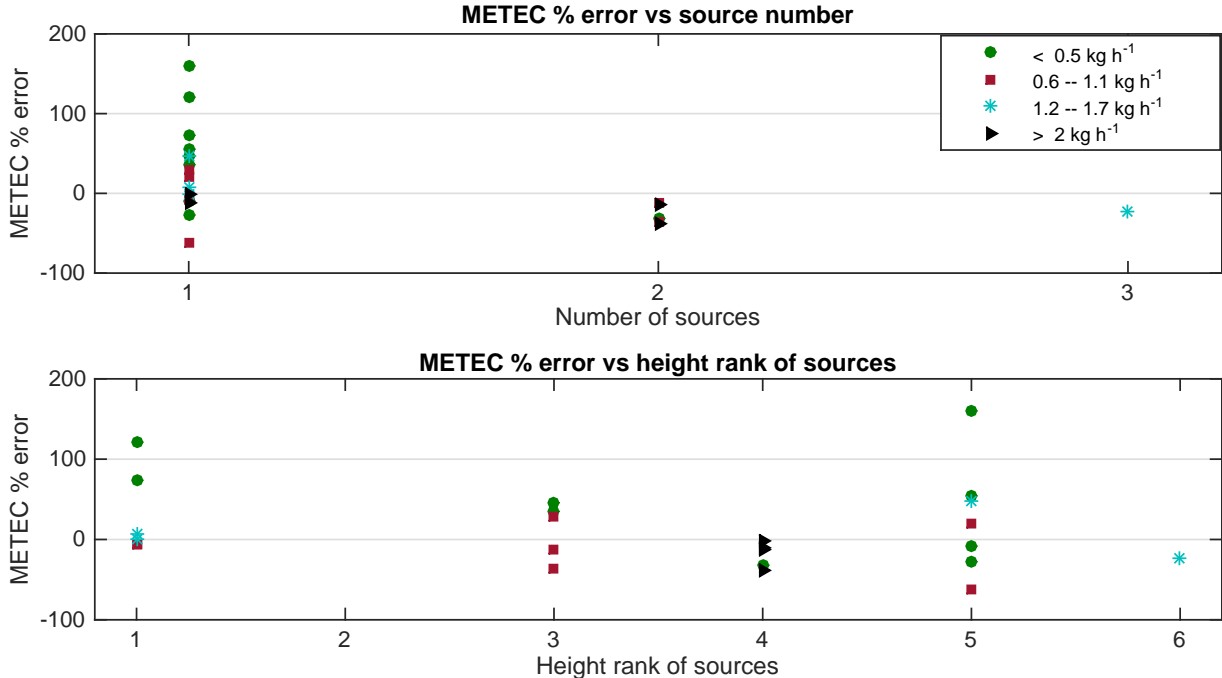

**Figure 11.** Percent error of the METEC-TR ranked by number of sources and height rank of sources. Positive error is OTM 33A over-estimating the known release rate. Height rank 1 is a ground-level emission point, height rank 5 is the highest emission point. Height rank 6 has both low, medium, and high release points. Icons indicate the size bin of the known release rate.



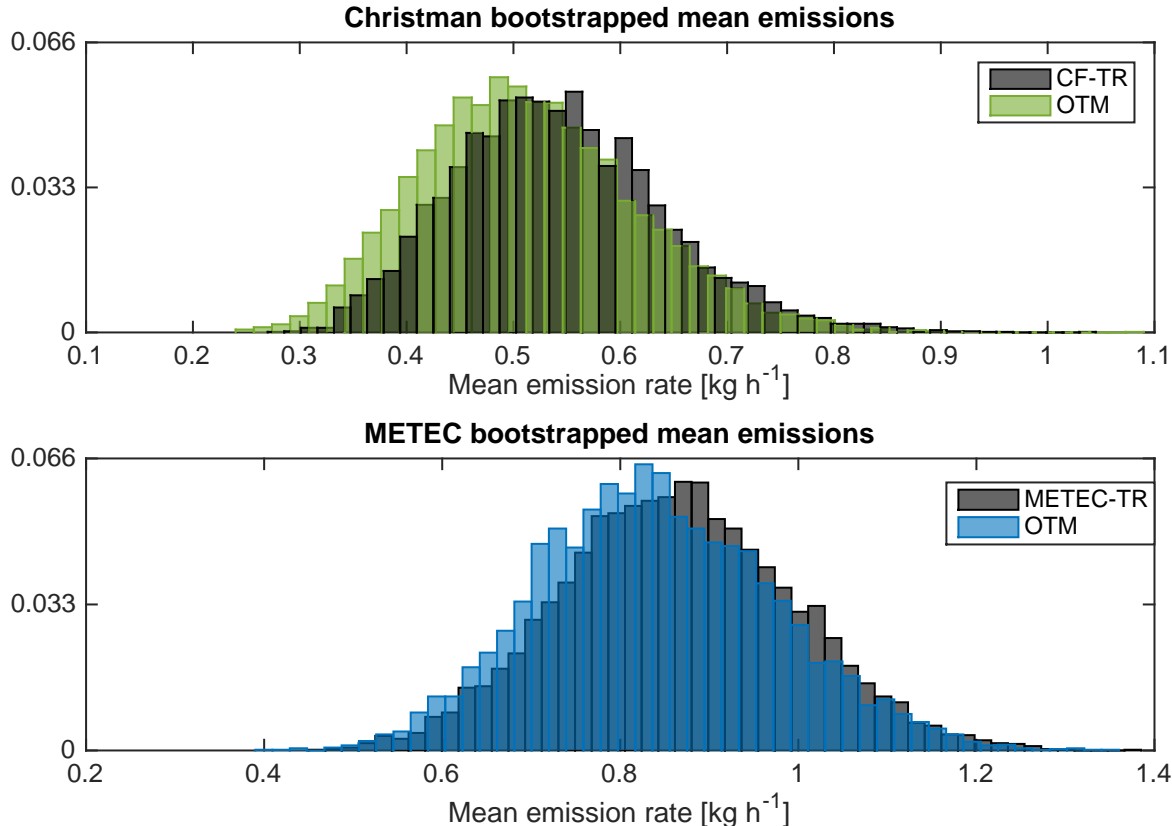

**Figure 12.** Probability Density Function (PDF) of bootstrapped mean mass emission flux for OTM measurements and the known releases. Mean and 95% CI in kg hr[-1] are as follows: Christman: known release - (0.54 (0.37,0.75), OTM - (0.51 (0.34,0.73)). METEC: known release - (0.85 (0.58,1.13)), OTM - (0.84 (0.60, 1.11)).



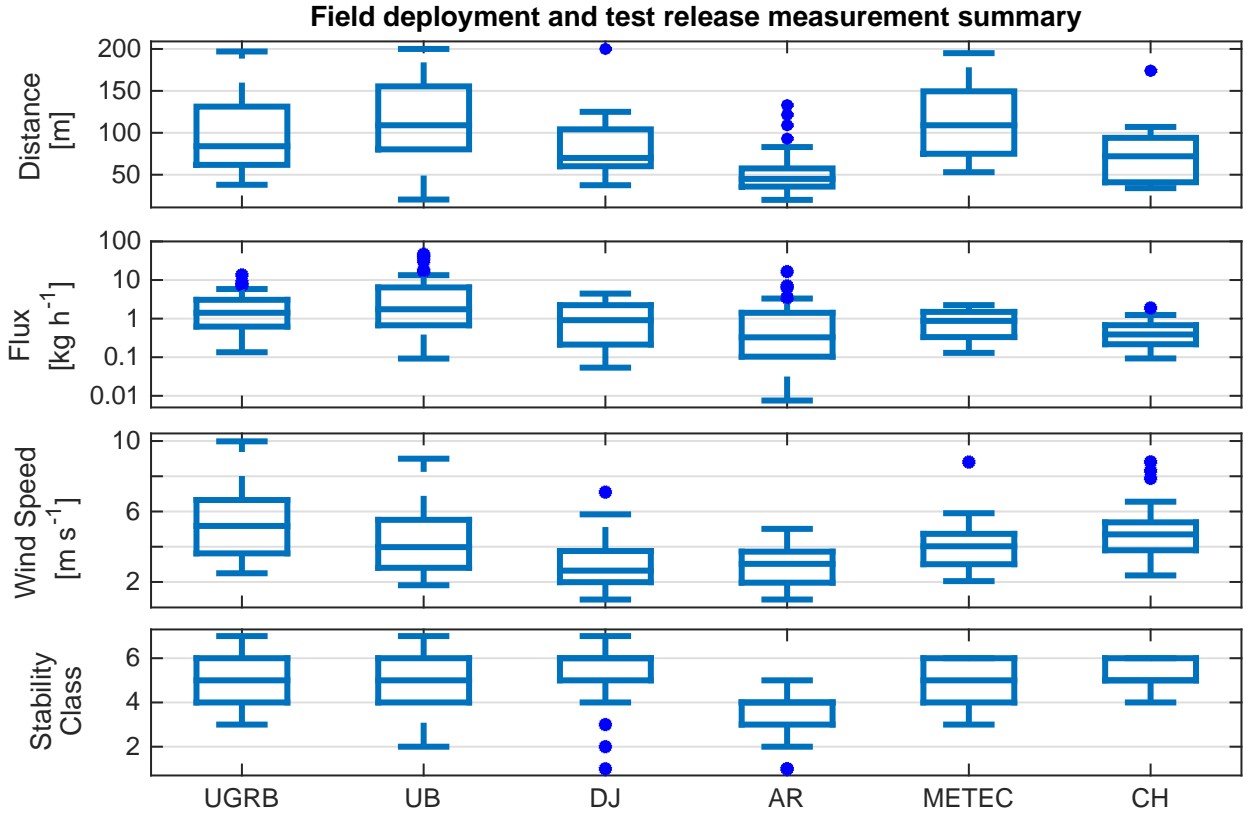

**Figure 13.** Summary of accepted OTM 33A measurements from field deployments and test releases. Basins from Robertson et al. 2017. Upper Green River Basin, Wyoming (UGRB), Uintah Basin, UT (UB), Denver-Julesburg Basin, CO (DJ), Fayetteville, Arkansas (AR).