# Peer review of "Constraining the Accuracy of Flux Estimates Using OTM 33A"

_Atmospheric Measurement Techniques, 2019_

## Referee Comment (RC1) · Anonymous Referee #1 · 15 Sep 2019

The EPA OTM 33A measurement technique is a mobile inspection method that can provide rapid assessment ($\sim$20 minutes) of whether a near-field, near-ground-level source is leaking and at what rate. The method has been widely used to detect and quantify methane emissions from oil and gas production well sites. The method was originally submitted by EPA's Office of Research and Development for inclusion in the Other Test Category (OTM) and is currently in draft form. Several researchers, including EPA's ORD, have previously performed controlled release tests involving single point-source releases to assess the performance of OTM 33A. This study expands on these previous tests by assessing OTM 33A performance under more realistic conditions using a faux oil and gas well site with multiple leak sources from typical well pad equipment. Since the most commonly used OTM 33A emission rate quantification ap-

proach (i.e., the point source Gaussian) assumes all emissions from a site converge to a point source, the use of a more realistic test environment with multiple sources provides a means to test the limits of this assumption. The authors' conclusion that, under this more realistic test conditions, OTM 33A has a "small but statistically insignificant low bias" and "does not drastically underestimate total emissions for an ensemble or group of measurements," is supported by the data and the analysis presented here. The paper is well written and the subject matter addressed here is important. However, the authors should consider providing additional details before the paper can be accepted for publication. In particular, the section describing OTM 33A sensitivity to source distances needs to be revised and clarified. Specific comments are provided below.

Page 2, line 13 to 14. Please expand on or provide a specific reference for the statement that VOC-rich emission sources are difficult to measure with onsite techniques.

Page 2, line 25 to 26. This sentence combines tracer flux method limitations (e.g., measurement distances) with method disadvantages (e.g., tracer flux techniques often require more implementation time than OTM 33A). It might be useful to distinguish between the two. Also, please provide a specific reference for the method limitations/requirements.

Page 4, Section 2.3. The OTM 33A emission rate quantification approach (the point source Gaussian) presented in this section is one of many possible quantification methods for OTM 33A. Other techniques (e.g., backward Lagrangian stochastic models) may have different performances than the PSG approach utilized here.

Page 4, lines 21 to 22. Please note that EPA considers the method to be more broadly applicable (i.e., not just for emission detection and quantification at point sources). EPA specifically identifies three source assessment modes for OTM 33A: (i) concentration mapping, (ii) source characterization and (iii) emission rate quantification.

Page 6, Section 3.1. The description of the OTM 33A test releases should be in the

[Figure]

Methods section. Similarly, the Methods section should include an overview of statistical tests performed, which are described in later sections under Results.

Page 6, lines 17 to 18. Please spell out how many "multiple release points" there were.

Page 6, lines 21 to 22. What informed the choice for the emission range tested here? Were the authors limited to this range? This has potential implications for how broadly applicable the results are, especially when larger emission rates (beyond the $\sim$ 2kg/h rate) are encountered in the field.

Page 6, lines 23 to 24. Did the authors perform tests at different source-to-observation distance configurations? If so, it would be helpful to provide a range/basic statistics here. Additional comment on this below.

Page 9, Section 3.3.1. This is an important section. Unfortunately, important details are missing. What was the average source distance for all test releases and how does this compare to the average in the Bell et al. study and in the EPA test? The data is shown in Figure 13, but it would be helpful to describe it here. Were there any measurements that were repeated at different source-to-observation distances to test OTM 33A sensitivity to source distances?

The second paragraph also needs more clarity. There is ambiguity in how the % changes in source distances were calculated. The % change could be based on (i) measurements of an emission source(s) at different observation distances, which means several 20-min samples of one known release were measured at different observation distances spanning a range of 20 m to 200 m, or (ii) fixed observation location, but the source-to-observation distance is varied (post-measurement) based on whether one assumes an average distance for all onsite sources or distance from a single known point source onsite. In the latter scenario, the difference in source distance would be no more than 6 m, the maximum separation distance for the multiple sources onsite (page 6, lines 17—18), which is small in the context of the 20—200 m range. Please provide more details in this section to help the reader understand

how the variability in source distances was assessed. Also, a plot showing the OTM % error as a function of source distance (similar to Figure 10 for wind speed) may help illustrate the point.

Page 9, Section 3.3.3. It is not clear here whether the sampling probe height was fixed or adjusted for different measurements. In Section 2.1 the sample inlet on the mobile laboratory is described to be located "4 meters" above the ground. And source heights are described to vary from 0—3 meters above the ground (page 6, lines 17—18). In testing OTM 33A sensitivity to source heights, were there specific configurations where one or more source heights was/were greater than the sampling probe height?

Page 10, lines 13 to 14. This is partly correct. The Alvarez et al. study also used other datasets obtained using other measurement techniques, not just the OTM 33A measurements in the four O&G basins described here.

Page 16, Figure 2. Please increase the font size for both figures (on all axis labels, legend and tick labels)

Page 25, Figure 11. It is not clear what heights (or range of heights) correspond to the height ranks shown here. What is the highest emission point? What are low, medium and high release points?

Page 27, Figure 13. It may be helpful to add a vertical line separating the dataset for actual OTM 33A measurements and OTM 33A release trials.

General comment on all figures: some figures have figure titles and others do not. Please review AMT guidelines and revise accordingly.
* * *

---

## Referee Comment (RC2) · Anonymous Referee #2 · 17 Oct 2019

This paper deals with a commonly used ground-based method (OTM33A) for estimating emissions rates. Recent papers have highlighted the relevance of site-level (facility-wide) emission estimates. The authors perform tests to assess accuracy of this approach in the context of methane emissions from single sites as well as ensembles (i.e., characterize emissions distributions from a population of sites). The results are relevant due to the increasing use of the approach. I recommend publication after some minor edits/clarification.

Two main points to be addressed/expanded by the authors: (1) Effect of multiple sources-distance selection (2) Determination of non-detects and potential effect of overestimation in determining fraction of sites that fall below detection limit.

Additional comments: INTRODUCTION: Page 1, Line 23: "Site-level measurements

are therefore necessary for improving emission estimates of the O&G production sector." This is true, might be also useful to mention importance of site-level measurements in conjunction with component-level measurements to understand source of emissions.

Page 2, Line 3: 'However, more permanent approaches are still under development and must be approved as equivalent monitoring technologies before they can replace existing EPA approved Leak Detection and Repair (LDAR) methods like optical gas imaging (OGI).' Suggest expanding discussion of difference between leak detection and leak (emissions) quantification, which is important in the context of LDAR and equivalency. One could argue that main goal of LDAR is not improving inventories, but repairing leaks. I think this idea needs to be further developed to link it to importance of site-level measurements.

METHODS It might be useful to briefly discuss the detection limit of the method (threshold for considering a site as non-detect). This is discussed in previous papers, but might be useful to summarize here. Consequently, discuss the potential overestimation at lower emission rates with the threshold for non-detects. This is something that matters for the ensemble.

Page 5, line 23: Might be good to mention that this could also affect flares (in addition to liquids unloadings).

Page 9, line 11-14. What happens with multiple sources on site? This paragraph hints at the importance of using OGI to locate source. Might be useful to expand on distance selection under various sources (i.e., based on highest emission point?)

Page 10, line 9-11. 'These results also indicate OTM 33A does not drastically underestimate the total emissions for an ensemble or group of measurements, and that scaling-up mean emissions measured with OTM 33A to an entire basin is a valid approach." This is an important conclusion from the paper since the ensemble is a common application of this method. Might be good idea to further highlight in the abstract.

Figure1: It might be useful to expand caption to include label of release points (i.e., what is the source of emissions). Figure 2: Significant figures for R parameter.

[Figure]

---

## Author Comment (AC1) · 21 Nov 2019

**Author responses to Anonymous Referee #1 are in bold below.**

The EPA OTM 33A measurement technique is a mobile inspection method that can provide rapid assessment (~20 minutes) of whether a near-field, near-ground-level source is leaking and at what rate. The method has been widely used to detect and quantify methane emissions from oil and gas production well sites. The method was originally submitted by EPA's Office of Research and Development for inclusion in the Other Test Category (OTM) and is currently in draft form. Several researchers, including EPA's ORD, have previously performed controlled release tests involving single point-source releases to assess the performance of OTM 33A. This study expands on these previous tests by assessing OTM 33A performance under more realistic conditions using a faux oil and gas well site with multiple leak sources from typical well pad equipment. Since the most commonly used OTM 33A emission rate quantification approach (i.e., the point source Gaussian) assumes all emissions from a site converge to a point source, the use of a more realistic test environment with multiple sources provides a means to test the limits of this assumption. The authors' conclusion that, under this more realistic test conditions, OTM 33A has a "small but statistically insignificant low bias" and "does not drastically underestimate total emissions for an ensemble or group of measurements," is supported by the data and the analysis presented here. The paper is well written and the subject matter addressed here is important. However, the authors should consider providing additional details before the paper can be accepted for publication. In particular, the section describing OTM 33A sensitivity to source distances needs to be revised and clarified. Specific comments are provided below.

**The authors greatly appreciate the reviewer's carefully considered comments on the manuscript. We have modified the manuscript to address the comments raised by the reviewer and we believe it is much improved because of these adjustments.**

Page 2, line 13 to 14. Please expand on or provide a specific reference for the statement that VOC-rich emission sources are difficult to measure with onsite techniques.

**A reference to the study by Brantley et al., 2015, which found that high volume samplers could malfunction in VOC-rich emission streams, has been added.**

**P2, L14-17 now reads: "Drawbacks of onsite measurements include difficulty measuring volatile organic compound (VOC) rich emission sources Brantley et al., 2015, inability to reach all emission sources (such as the tops of free-standing tanks), difficulty measuring intermittent sources, and the time required for each inspection Brantley et al., 2014, Bell et al., 2017, Ravikumar et al., 2018."**

Page 2, line 25 to 26. This sentence combines tracer flux method limitations (e.g., measurement distances) with method disadvantages (e.g., tracer flux techniques often require more implementation time than OTM 33A). It might be useful to distinguish between the two. Also, please provide a specific reference for the method limitations/requirements.

**The tracer flux method limitations versus disadvantages have been separated. We now include a reference to the Yacovitch et al., 2017 study, which states the TFR method measured 2.5 sites/day. It is now explicitly stated that some of the tracers (acetylene is**

common) are flammable. This flammability requires many safety protocols for dealing with acetylene, including emergency shutoff valves and minimum wind speed requirements to prevent pooling near possible ignition sources.

**P2, L25-28 now read: "Limitations of TFR include the reliance on downwind roadways of sufficient distance (~0.5 - 2 km) and reliable wind direction (Omara et al., 2018; Roscioli et al., 2015). Drawbacks of using TFR to estimate methane emissions include the amount of time required to estimate emissions from one site (2.5–2.8 sites day-1) (Yacovitch et al., 2017), and the need to transport and release compressed tracer gases (some of which are flammable such as acetylene) near O&G facilities."**

Page 4, Section 2.3. The OTM 33A emission rate quantification approach (the point source Gaussian) presented in this section is one of many possible quantification methods for OTM 33A. Other techniques (e.g., backward Lagrangian stochastic models) may have different performances than the PSG approach utilized here.

**The reviewer is correct, the following text has been added to clarify, "While several quantification approaches are possible with OTM 33A, the one most commonly employed is an inverse Gaussian approach and this approach is the focus of this manuscript."**

**P4, L25-26 now reads: "While several quantification approaches are possible with OTM 33A, the one most commonly employed is an inverse Gaussian approach, which is the focus of this manuscript."**

Page 4, lines 21 to 22. Please note that EPA considers the method to be more broadly applicable (i.e., not just for emission detection and quantification at point sources).

**The following comment has been added to the text.**

**P4, Line 23 now reads: "OTM 33A is one of the EPA Geospatial Measurement of Air Pollution Remote Emission Quantification (GMAP-REQ) techniques that was designed to observe, characterize, and/or quantify emissions from a variety of sources, though OTM 33A has been used most to measure emissions from O&G operations.**

EPA specifically identifies three source assessment modes for OTM 33A: (i) concentration mapping, (ii) source characterization and (iii) emission rate quantification.

**The text has been corrected to reflect the three assessment modes.**
**P4, Line 26-27 now reads: "OTM 33A has three operational parts: concentration mapping, source characterization, and emission rate quantification."**
**We have also added an additional sentence describing source characterization to P4, L29 - 30 which reads: "Source characterization includes observations of temporal variability and emissions composition. If enhancements of methane or other trace gases are detected during downwind transects of a possible source, the laboratory is parked 20–200 m directly downwind within the emission plume to quantify emissions."**

Page 6, Section 3.1. The description of the OTM 33A test releases should be in Methods section. Similarly, the Methods section should include an overview of statistical tests performed, which are described in later sections under Results.

**Section 3.1 has been moved to the methods. While description of the statistical tests would typically go in the Methods section, for this manuscript we feel it is appropriate to leave it in the results because the potential bias of certain statistical analyses is a key result.**

Page 6, lines 17 to 18. Please spell out how many "multiple release points" there were.

**Information regarding the number of unique release points used for the METEC test releases has been added.**
**P6, L now reads: "For this study, we used one METEC site representative of a small O&G facility that included a condensate storage tank, separator, and well head, all of which were plumbed to be possible emission sources, 11 of which were used in this study(Fig. 1). This resulted in 15 release configurations that had from 1–3 release points at different heights (0.33–4 meters), up to 6 meters apart from one another."**

**The caption for Figure 1 has also been adjusted to read: "METEC facility with nine of the 11 release points circled. Release points include (clockwise from top of tank) tank candy cane, tank thief hatch, tank front flange, wellhead Kimray packing, wellhead hand valve packing, separator burner fuel supply, separator Kimray vent, separator PRV, and separator house PRV. Not pictured: wellhead lubricator flange and wellhead pressure gauge. The UW mobile laboratory is in the background."**

Page 6, lines 21 to 22. What informed the choice for the emission range tested here? Were the authors limited to this range? This has potential implications for how broadly applicable the results are, especially when larger emission rates (beyond the $\sim$ 2kg/h rate) are encountered in the field.

**The release rate range was constrained by the facilities/test release configurations. We would have liked to measure larger release rates, but this was not possible. This information, as well as the range of bootstrapped mean emission rates for four basins and a discussion of the limitations of this measurement range have been added to the Conclusions.**

**P12, L25-29 : "For both test release experiments, the maximum release rates (2-2.15 kg h$^{-1}$) were constrained by available resources and facility throughput and, while they represent a large fraction of emission rates observed in the field, they do not fully encompass the dynamic range of emissions observed in an O&G basin. The bootstrapped mean emission rates from four O&G basins measured by the University of Wyoming range from 0.68–3.7 kg h-1(Robertson et al., 2017), suggesting the range of these test releases may not be representative of the largest emission rates observed in the field (Fig. 13)."**

Page 6, lines 23 to 24. Did the authors perform tests at different source-to-observation distance configurations? If so, it would be helpful to provide a range/basic statistics here. Additional comment on this below.

**The mean and range of measurement distances for each test release experiment has been added to their descriptions.**
**P6, L16-17: "Mean measurement distance was 78 m, with a range of 34–174 m."**
**P6, L29: "Mean measurement distance was 114 m with a range of 53–195 m."**

**Additionally, an excel workbook giving the distance to source, release rate, release height, average wind speed, and OTM estimated emission rate has been added to the SI for both test release experiments.**

Page 9, Section 3.3.1. This is an important section. Unfortunately, important details are missing. What was the average source distance for all test releases and how does this compare to the average in the Bell et al. study and in the EPA test? The data is shown in Figure 13, but it would be helpful to describe it here. Were there any measurements that were repeated at different source-to-observation distances to test OTM 33A sensitivity to source distances?

**Section 3.3.1 (now Section 3.2.1) now includes a summary of average and range of source distance for the test releases and the Arkansas data. We could find no reference for the distance observed for the complete set of EPA test releases (N = 107), but the preliminary report on OTM 33A by Thoma et al., 2012 included 24 test releases with measurement distances ranging from 18-103m.**

**P10, L9-16 now read: "OTM 33A sensitivity to distance was also tested in the field during the METEC test releases. For configurations that had both a "closer" (generally<70 m) and "farther" (generally>100m) measurement distance for replicate measurements, the closer measurement had a flux estimate closer to the known release 78% of the time (SI Sect. 1.4). The average distance of the closer replicate measurements (78 m) is comparable to the average measurement distances for the CF-TR of 78 m, smaller than the mean METEC-TR distance of 114 m, and larger than the measurement distances during the Arkansas campaign of 46 m (20–113 m) (Robertson et al., 2017; Bell et al., 2017). For both the CF-TR and the METEC-TR, there is no obvious increase in% error as measurement distance increases (Fig. 10(a)), suggesting the underestimation reported by Bell et al. cannot be blamed solely on closer measurement distances."**

**The METEC test release were designed to measure the same release configuration (release points and release rate) two to three times. Unfortunately, only 10 of the 15 attempted configurations had duplicate measurements that passed the DQI. Initially, we felt the sample size was too small to include statistical tests for this sub-set of data, but we have added this information to Section 3.1.1, 3.2.1, and SI Sect. 1.4.**

**Lastly, the mean statistics for measured basins and test releases have been added to the caption of Figure 13.**

The caption now reads: "Figure 13. Summary of accepted OTM 33A measurements from field deployments and test releases (right of vertical line). Basins from Robertson et al. 2017. Upper Green River Basin, Wyoming (UGRB), Uintah Basin, UT (UB), Denver-Julesburg Basin, CO (DJ), Fayetteville, Arkansas (AR). Mean statistics (from left to right) are as follows. Distance [m]: 98, 114, 83, 51, 114, 78. Flux [kg h-1]: 2.41, 6.99, 1.51, 1.27,0.51, 0.96. Mean wind speed [m s-1]: 5.3, 4.2, 3.1, 2.9, 4.1, 4.9. Stability class: 5.0, 4.9, 5.3, 3.5, 5.0, 5.4."

The second paragraph also needs more clarity. There is ambiguity in how the % changes in source distances were calculated. The % change could be based on (i) measurements of an emission source(s) at different observation distances, which means several 20-min samples of one known release were measured at different observation distances spanning a range of 20 m to 200 m, or (ii) fixed observation location, but the source-to-observation distance is varied (post-measurement) based on whether one assumes an average distance for all onsite sources or distance from a single known point source onsite. In the latter scenario, the difference in source distance would be no more than 6 m, the maximum separation distance for the multiple sources onsite (page 6, lines 17âˇT18), which is small in the context of the 20âˇAˇT200ˇm range. Please provide more details in this section to help the reader understand how the variability in source distances was assessed. Also, a plot showing the OTM % error as a function of source distance (similar to Figure 10 for wind speed) may help illustrate the point.

**More clarification has been added to Section 3.2.1 to distinguish between the two potential source distance errors identified by the reviewer.**

**Beginning on P9, L25: "OTM 33A sensitivity to source distance was tested two ways for the METEC test releases. The following test was performed during the data analysis stage, and compared the flux estimated using the average distance of all the components that could be sighted with the range finder from the van (e.g. wellhead, separator, tank) to the flux estimated using the distance to the known release point or point distance (identified using the FLIR camera). Although the well pad measured at the METEC facility was quite small (∼6 m by 6 m), the average source distance was larger than the specific source distance∼60% of the time. The change in the OTM 33A flux (ΔFlux) as a result of changing the measurement distance (ΔDistance) was found using Equations 3 and 4.**

**ΔDistance=(Average Distance−Point Distance)/(Average Distance) × 100         (3)**
**% ΔFlux=(Average OTM−Point OTM)/Average OTM × 100         (4)**

**A correlation plot of %ΔDistance and %ΔFlux suggests that for a 5% change in source distance, the OTM 33A flux estimate would increase by almost 10% (Fig. 9(a)). In terms of mass error, the OTM flux estimated by the average or specific source distance has very little impact in the over- or under-estimation of the METEC known release (Fig. 9(b)). Allowing this fit to have an intercept changes the linear fit toy= 0.978x−0.03, a negligible difference. Source distance related error is small in the context of the ±70% measurement error, but this analysis underscores how determination of the exact emission point can further reduce errors in the field."**

**Additionally, a panel has been added to Figure 10 to show how estimated fluxes varied versus distance of the mobile lab from the emission source.**

Page 9, Section 3.3.3. It is not clear here whether the sampling probe height was fixed or adjusted for different measurements. In Section 2.1 the sample inlet on the mobile laboratory is described to be located "4 meters" above the ground. And source heights are described to vary from 0ǎ˘T3 meters above the ground (page 6, lines 17â ˘ A˘T18). In ˘ testing OTM 33A sensitivity to source heights, were there specific configurations where one or more source heights was/were greater than the sampling probe height?
**A better description of the mast has been added to Section 2.1 to emphasize that it remains at a fixed height of 4 m.**

**P4, L5-7 now read: "The University of Wyoming mobile laboratory is a customized Freightliner Sprinter van. The front of the van is equipped with a horizontal mast that projects instrumentation and the inlet at a fixed height of 4 meters above the ground slightly beyond the vehicle's front bumper."**

**The METEC facility had release points ranging from 0.33-4.33 meters above ground level.**

**P6, L23-24 now read: "This resulted in 15 release configurations that had from 1–3 release points at different heights (0.33–4 meters), up to 6 meters apart from one another."**

Page 10, lines 13 to 14. This is partly correct. The Alvarez et al. study also used other datasets obtained using other measurement techniques, not just the OTM 33A measurements in the four O&G basins described here.
**This wording has been corrected. It now states that data from these basins, "represented a significant fraction of data, along with other field campaigns."**

Page 16, Figure 2. Please increase the font size for both figures (on all axis labels, legend and tick labels)
**Font size has been increased.**

Page 25, Figure 11. It is not clear what heights (or range of heights) correspond to the height ranks shown here. What is the highest emission point? What are low, medium and high release points?

**Figure 11 has been remade to show the average height of the emission sources in meters instead of by rank.**

Page 27, Figure 13. It may be helpful to add a vertical line separating the dataset for actual OTM 33A measurements and OTM 33A release trials.

**The suggested change has been made.**

General comment on all figures: some figures have figure titles and others do not. Please review AMT guidelines and revise accordingly.

**All figures now have figure titles.**

---

## Author Comment (AC2) · 21 Nov 2019

**Author responses to Anonymous Referee #2 are in bold below:**

This paper deals with a commonly used ground-based method (OTM33A) for estimating emissions rates. Recent papers have highlighted the relevance of site-level (facility-wide) emission estimates. The authors perform tests to assess accuracy of this approach in the context of methane emissions from single sites as well as ensembles (i.e., characterize emissions distributions from a population of sites). The results are relevant due to the increasing use of the approach. I recommend publication after some minor edits/clarification. Two main points to be addressed/expanded by the authors: (1) Effect of multiple sources-distance selection (2) Determination of non-detects and potential effect of overestimation in determining fraction of sites that fall below detection limit. Additional comments:

**We greatly appreciate the reviewer's careful consideration of the manuscript. We have addressed the noted issues as detailed below.**

INTRODUCTION: Page 1, Line 23: "Site-level measurements are therefore necessary for improving emission estimates of the O&G production sector." This is true, might be also useful to mention importance of site-level measurements in conjunction with component-level measurements to understand source of emissions.

**The text on line 23 has been changed to include component-level measurements and the study by Brandt et al., 2014, is included to help emphasize that point.**

**P1, L23-24 now read: "Site- and component-level measurements are therefore necessary for improving emission estimates of the O&G production sector (Brandt et al.,2014).**

Page 2, Line 3: 'However, more permanent approaches are still under development and must be approved as equivalent monitoring technologies before they can replace existing EPA approved Leak Detection and Repair (LDAR) methods like optical gas imaging (OGI).' Suggest expanding discussion of difference between leak detection and leak (emissions) quantification, which is important in the context of LDAR and equivalency. One could argue that main goal of LDAR is not improving inventories, but repairing leaks. I think this idea needs to be further developed to link it to importance of site-level measurements.

**Additional text has been added to page in an attempt to emphasize that LDAR does not typically generate data that can be used to improve emission inventories. While we agree that there is ample material to be discussed in terms of LDAR methods and equivalency, we believe this is beyond the scope of the current manuscript.**

**P2, L6-10 now read: "Annual or semi-annual LDAR programs already in place rarely quantify total emissions from a site, and the efficacy of these programs depends on many factors including employee experience, leak size, and meteorological variables like wind speed and temperature (Ravikumar et al., 2016, 2018). This makes LDAR programs an important tool for finding leaks and reducing emissions, but they often do not explicitly quantify or provide data of the actual emission rate from production sites, and this limits usefulness for improving emission inventories."**

METHODS It might be useful to briefly discuss the detection limit of the method (threshold for considering a site as non-detect). This is discussed in previous papers, but might be useful to summarize here. Consequently, discuss the potential overestimation at lower emission rates with the threshold for non-detects. This is something that matters for the ensemble.

**The method limit of detection has been added to the methods section.**
**P5, L30-31 now reads: "The estimated lower detection limit of the method is 0.01 g s$^{-1}$ 0.036 kg h$^{-1}$) (Brantley et al., 2014)."**

**In the current study there were no "non-detects" meaning there is no bias in the Christman or METEC ensembles. In the field, careful consideration of non-detects is essential, but we feel this is best addressed in the papers covering those field deployments as methodology varies slightly from one study to another. Overall, the slight underestimation of total mass flux found in this study and the large underestimation reported by Bell et al., 2017 support OTM 33A being, if anything, slightly low for an ensemble of measurements. In general, it is not too critical what number is inserted for the low emission wells as the mean of the ensemble is dominated by higher emission sites and the uncertainty in the number of high emission sites.**

Page 5, line 23: Might be good to mention that this could also affect flares (in addition to liquids unloadings).

**P5, L27-29 now read: "OTM 33Astruggles to quantify plumes with a particularly high vertical velocity or buoyancy (such as manual unloadings, lit or unlit flares, or very hot emissions)."**

Page 9, line 11-14. What happens with multiple sources on site? This paragraph hints at the importance of using OGI to locate source. Might be useful to expand on distance selection under various sources (i.e., based on highest emission point?)

**The analysis presented here suggests that, at least for emission points that are within 6 m of each other, no selection of a specific source is necessary given that the observed error of ~10% is much smaller than other errors associated with the method. This section has been expanded to more clearly explain the relatively small impact of not knowing the exact source location on smaller sites (these are the sites typically measured via OTM 33A).**

Page 10, line 9-11. 'These results also indicate OTM 33A does not drastically underestimate the total emissions for an ensemble or group of measurements, and that scaling up mean emissions measured with OTM 33A to an entire basin is a valid approach." This is an important conclusion from the paper since the ensemble is a common application of this method. Might be good idea to further highlight in the abstract.

**We agree this is an important finding. We believe the statement in the abstract on Page 1 Lines 12-13 that, "an ensemble of OTM 33A measurements may have a small but**

**statistically insignificant low bias." makes this point without overstating what can be determined from the current study.**

Figure1: It might be useful to expand caption to include label of release points (i.e., what is the source of emissions).
**The caption of Figure 1 has been expanded to include descriptions of all of the pictured release points, as well as the total number of release points (11).**

Figure 2: Significant figures for R parameter.
**Significant figures for all R parameter values have been appropriately reduced.**